# Modeling of Microneedle Arrays in Transdermal Drug Delivery Applications

**DOI:** 10.3390/pharmaceutics15020358

**Published:** 2023-01-20

**Authors:** Francisco Henriquez, Diego Celentano, Marcela Vega, Gonzalo Pincheira, J. O. Morales-Ferreiro

**Affiliations:** 1Facultad de Ingeniería, Departamento de Tecnologías Industriales, Universidad de Talca, Camino Los Niches Km 1, Curicó 3340000, Chile; 2Departamento de Ingeniería Mecánica y Metalúrgica, Centro de Investigación en Nanotecnología y Materiales Avanzados (CIEN-UC), Millennium Institute on Green Ammonia as Energy Vector (MIGA), Pontificia Universidad Católica de Chile, Av. Vicuña Mackenna 4860, Macúl, Santiago 8331150, Chile; 3Instituto de Investigación Interdisciplinaria, Vicerrectoría Académica, Universidad de Talca, 1 Poniente 1141, Talca 3460000, Chile

**Keywords:** CFD, design, interstitial pressure, lidocaine, microneedles

## Abstract

The use of computational tools for the development of technologies in fields such as medicine and engineering has facilitated the process of designing new components and devices for these areas. In this work, two proposals focused on a hollow microneedle array (MNA) for the administration of an analgesic drug are shown and evaluated by means of a computational fluid dynamics (CFD) simulation distributed in three stages. In the first stage, the behavior of lidocaine through the MNA was evaluated as a workflow. Then, the possible entry of the drug into the organism, which was established as a porous aqueous medium, was modeled. Finally, a joint simulation was performed to understand the general behavior in the interaction between the outflow of an MNA and the body to which lidocaine is administered. The input parameters to the simulation were set at a velocity of 0.05 m∙s^−1^, at a pressure of 2000 Pa, the dominant behavior was defined as laminar flow, and a resistive pressure at the inlet of 400 Pa. Our results indicate that the vertical flow exhibits a better fluid distribution across the MNAs and favorable infiltration behavior, representing better delivery of the analgesic to the skin capillaries.

## 1. Introduction

Transdermal drug delivery systems (TDDS) based on microneedles (MNs) have been studied in recent years for their painless and skin-friendly performance and pharmacological advantages [1,2,3,4]. The principal disadvantage of this device is the low volume of dosing. The main purpose of these MNs is to break the external barrier of human skin to allow entry of drugs, antigens, or proteins into the blood system; to take biological samples; or as both therapeutic and cosmetic treatments [4,5,6,7]. This mode of action takes advantage of transcellular pathways for the drug to make its way through the various layers of skin into the blood capillaries [1].

The MNs can be classified according to different criteria, the most widely used being the drug release mechanism, as: solids, coated, dissolvable, hollow, and hydrogels [2,3]. The MNs are design according to a given application, where the geometry is a key parameter and largely defines the functionality in cosmetic, therapeutic, or clinical fields [3,8]. In this way, manufacturing processes can offer characteristic geometries attributable to the manufacturing method as is the case with laser ablation or air-droplet blowing [2,3,4].

Thus, the use of MN-based devices has presented a series of results in different works and with different drugs, which have been used as vaccines for the treatment of cancer, diabetes, influenza, and pain, among other medical conditions [3,5,9]. Table 1 shows a summary of some studies attributable to the use of MNs.

Among the types of MNs, hollow MNs stand out, offering an alternative to open a continuous supply channel. However, their hollow and thin characteristics present a challenge at the structural level [8,10,11,12]. In addition, the behavior of the fluid passing through the system and the composition of the transported drugs must be considered [13], as this could determine the performance of a specific device [10]. It is worth mentioning that it is necessary to understand the behavior of the drug in the organism in order to improve the efficiency of administration and the response times to a given dose. In this way, it is necessary to know the mechanical properties of the fluid to work with, in order to establish specific working conditions such as temperatures, concentrations, doses to handle, storage times, etc.

Given the above and according to the microscale nature involved in this problem, the specific characteristics of microfluidics must be considered which, in recent years, have been supported by computational tools, such as computational fluid dynamics (CFD) to model the behavior of fluids in areas such as engineering, medicine, and pharmaceuticals. In this context, the objective is to simulate and understand how microfluids are affected in specific conditions and give various answers in different types of research [10,14,15,16].

The present work aims to show the behavior of lidocaine, an analgesic fluid, through an MNA, and to determine how the fluid would be distributed through the transcellular pathways of the skin using CFD simulation tools, as presented in Figure 1. In particular, two MNA-based designs are analyzed, which represent an internal lateral flow and an internal vertical flow. The flow distribution both in the MNA and in the porous medium are considered separately, in order to analyze the homogeneity in the distribution of the analgesic to each MN, to finally simulate the behavior of the joint situation, to assess the effect of the transition between the MNA and the organism, and how the resistive pressure of the organism influences the entry of lidocaine. These considerations allow to evaluate a more realistic interaction in the transition of the fluid from one system to another with a lower associated uncertainty.

## 2. Materials and Methods

### 2.1. Materials

#### 2.1.1. Analgesic Drug

Among the common anesthetics used in medicine, lidocaine and bupivacaine are normally applied intradermally and both are used as agents that block peripheral nerves and relieve focused pain [17,18]. Lidocaine, depending on its form of administration, can have different problems, ranging from systemic side effects in its intravenous administration [19], to discomfort in the oral intake of pills, especially in the geriatric and pediatric population [20]. This tends to be a major challenge in medical issues and the topical alternative can present an unacceptable response time in many cases [9,21].

From previous studies on the experimental properties of lidocaine hydrochloride in aqueous solution, it is shown that at a temperature of 298.15 K and a concentration of 0.00978 mol·kg^−1^, a density of 997,944 kg·m^−3^ and a molarity of 0.02578 mol·kg^−1^ are obtained, plus an apparent molar volume of 236.27 m^3^·mol^−1^ for a pressure of 1 atm [17]. The analysis was accompanied by a computational simulation based on CFD that closely approximates the behavior of lidocaine in specific situations.

#### 2.1.2. Microneedle Design

For the MNA design, a hollow microneedle was considered based on an eccentric cone with a height of 600 µm, a basal diameter of 300 µm, and a microchannel of 90 µm in diameter. Those dimensions were found to provide structural stability when breaking the external barrier of human skin [8]. Due to its structural resistance, there is an array of 4 × 4 microneedles with internal microchannels, which seek, through geometric parameters, to ensure a homogeneous distribution of the fluid throughout the array. Therefore, two fluid distribution designs were proposed, a lateral flow (LF) and a vertical flow (VF), as shown in Figure 2 and Figure 3, respectively.

For the LF design, a system of rectangular microchannels 100 µm high by 90 µm wide was drawn, this according to the geometric parameters of the hollow MN, which form a branch system whose purpose is to propose a stable distribution of the analgesic along the MNA (Figure 2).

For the VF design, a more robust configuration is observed, which presents microchannels with a more complex geometry when presented as a distribution that uses the three axes as flow directions as shown in Figure 3. This design, like the previous one, aims to achieve a stable distribution based on a symmetrical design.

The epidermis was established as a porous medium system, where the output geometry of the MNA was the input geometry.

Finally, the innermost layers of the skin were defined as porous media, which adopted the specific geometry of the MN considered for this work. With this, the most favorable situation of effective penetration is proposed, leaving the epidermis as shown in Figure 4.

### 2.2. Methods

#### 2.2.1. Mathematical Modeling

To define the behavior of the fluid through the proposed MNAs, it is required to solve a series of equations based on continuity, momentum, diffusion and porous media. In this way, the ANSYS R1 software [22], using the finite element method (FEM) and CFD, proposes some equations shown below.

##### Momentum and Continuity

For incompressible fluids, the conservation of mass formulation is given by:(1)$$\frac{\partial \rho }{\partial t}+\nabla \cdot \left(\rho \bar{v}\right)=0$$

Then, the conservation of momentum is given as:(2)$$\frac{\partial }{\partial t}\left(\rho \bar{v}\right)+\nabla \cdot \left(\rho \bar{v}\bar{v}\right)=-\nabla p+\nabla \left(\bar{\tau }\right)+\rho \bar{g}+\bar{F}$$
where *p* is the pressure, $\bar{\tau }$ is the stress tensor, *ρ* is the fluid density, $\rho \bar{g}$ is the gravitational body force, and $\bar{F}$ is the external body force.

The stress tensor is given by:(3)$$\bar{\tau }=\mu \left(\bar{\nabla v}+\nabla {\bar{v}}^{T}\right)$$
where *µ* is the viscosity of the fluid.

##### Mass Transfer

Mass transfer by diffusion was modeled according to Fick’s second:(4)$$\frac{\partial C}{\partial t}+\bar{u}\cdot \nabla C=D\nabla^{2}C$$
where *C* is the concentration and *D* is the diffusivity

##### Porous Media

Porous media were modeled by the addition of a momentum source term to the standard fluid flow equations. The source term is composed of two parts: a viscous loss term and an inertial loss term:(5)$$S_{i}=-\left(\overset{3}{\underset{j=1}{\sum }}D_{ij}\mu v_{j}+\overset{3}{\underset{j=1}{\sum }}C_{ij}\frac{1}{2}\rho \left|v\right|v_{j}\right)$$
where $S_{i}$ is the source term for the ith momentum equation, $\left|v\right|$ is the magnitude of the velocity, which contributes to the pressure gradient in the porous cell, creating a pressure drop that is proportional to the fluid velocity in the cell.

If the porosity is fixed as a homogeneous term throughout the domain, the source term is expressed as:(6)$$S_{i}=-\left(\frac{\mu }{\alpha }v_{i}+C_{2}\frac{1}{2}\rho \left|v\right|v_{i}\right)$$
where α is the permeability, $C_{2}$ inertial resistance factor, and i refers to the X, Y, or Z axes.

Then, for laminar flows in porous media, Darcy’s law is applied as:(7)$$\nabla p=-\frac{\mu }{\alpha }\bar{v}$$

Given the above, the pressure in each direction in the porous region is expressed as:(8)$$\Delta p_{i}=\overset{3}{\underset{j=1}{\sum }}-\frac{\mu }{\alpha_{ij}}{\bar{v}}_{j}\Delta n_{i}$$
where $\Delta n_{i}$ is the thickness of porous media in each axis, X, Y, or Z.

##### Interstitial Fluid

The formation of interstitial fluid depends on several factors, and it is responsible for maintaining an intracellular pressure, which represents an opposing force to the filtration of fluids towards the blood capillaries. The physicist E. Starling formulated an equation for net infiltration capacity for the absorption in blood vessels described as [23,24]:(9)$$J_{V}=L_{P}A\left[\left(P_{c}-P_{if}\right)-\sigma \left(COP_{c}-COP_{if}\right)\right]$$
where $J_{V}$ is the net infiltration capacity, $L_{P}$ is the capillary hydraulic conductivity, A is the surface area available for filtration, $P_{c}$ and $P_{if}$ are capillary and pore hydrostatic pressure respectively, σ is the capillary osmotic reflection coefficient, $COP_{c}$ and $COP_{if}$ are the capillary and interstitial colloidal pressure, respectively.

#### 2.2.2. Numerical Simulation

To verify the behavior of the fluid according to the assumptions presented above and the subsequent entry into the human body, it is necessary to define input parameters, such as mass flow rate, volumetric flow rate, inlet pressure, and fluid velocity input. With these parameters, through the use of CFD simulation, the output behavior for a specific flow rate is obtained through the corresponding modeling.

The CFD simulation was divided into three fundamental stages. The first stage consists of performing the simulations for the LF and VF design proposals and establishing the behavior of the analgesic flow through the MNAs and how the fluid is distributed to each of the MNs (see Figure 5 and Figure 6). The conditions applied in this stage were a speed of 0.05 m·s^−1^ at a pressure of 2000 Pa for the input [25]. In addition, laminar flow conditions were maintained throughout the domain, with inert walls and zero continuous outlet pressure.

The second stage sought to show the possible behavior of the fluid through a porous aqueous medium, a situation established to model the transcellular pathways of the epidermis and dermis, as shown in Figure 7. For this, within the simulation conditions, a condition of non-return flow is considered, due to the significant pressure difference between those of the analgesic and the resistive tissue. This condition mimics the real clinical application [26]. Moreover, a laminar behavior of the fluid in the defined geometry is assumed.

Finally, in the third stage, the analysis is extended to visualize the possible behavior experienced by the analgesic fluid in the transition from the MNA to the body, specifically in the epidermis and, in addition, to include the resistive pressure factor that the interstitial fluid could produce. In this way, the entry conditions to the array are maintained at a speed of 0.05 m·s^−1^ at a pressure of 2000 Pa, with the conditions inside the MNA and the porous media remain the same as in the previous sections and, finally, a resistance pressure of 400 Pa is included, established as normal transcellular pressure for healthy tissues [24].

All these simulations were carried out with the ANSYS FLUENT 2020 R1 software [22].

It is worth mentioning that the time windows considered in the modeling performed are in the order of 60 to 70 min [27] for doses that consider a developed state with conditions of 0.5 m·s^−1^ at 2000 Pa at the inlet of the MNA. The time step size used was 0.002 s.

## 3. Results and Discussion

The use of computational tools has made it possible to optimize designs by predicting specific situations through simulation based on recorded experimental data. To maintain greater clarity in specific situations, tools such as CFD have contributed to the prediction of fluid behavior and help establish operating conditions in specific designs [10,14,15,16]. The results corresponding to the stages previously described are separately presented below.

### 3.1. Simulation of the MNA

The results shown in Figure 5 and Figure 6 demonstrate the behavior of lidocaine as a working fluid according to the physical properties shown in Section 2. From these images, it is observed that the MNA VF (Figure 6) presents a better distribution of the fluid to each of the MNs, due to a more stable profile of velocities and pressures is presented throughout the array, which could represent a greater homogeneity in drug delivery. Moreover, the maximum velocity at the outlet in each of the MN arrays was 0.24 m∙s^−1^, with an inlet pressure to the microchannels of 360.3 Pa in the primary distribution section, which indicates a considerable drop in relation to the input pressure, and, at the same time, an increase in the delivery speed for the analgesic.

It should be noted that the distribution of the analgesic along the matrix is corroborated with previous studies in the field of CFD analysis [16,28,29].

### 3.2. Simulation of the Porous Media

To define the behavior that the analgesic fluid could have when entering the inner layers of the skin, a system of aqueous porous media was simulated. As presented in Figure 7, a uniform input with slight curves can be clearly observed before the input speed, which were set in the same way as in the previous stage. In addition, a stable pressure is shown throughout the studied domain, due to the simulation parameters which induce an outlet pressure equal to zero at the bottom of the presented geometry. It should be noted that the behavior obtained from the simulation assumed to the named working conditions, where the flow always maintains a constant output and there is no force that interrupts the fluid filtration paths.

### 3.3. Simulation of the System MNA—Porous Media

The body has a natural resistance to the entry of foreign elements formed by interstitial pressure. Along with this, it must be considered that the fluid may not have a stable behavior during the transition between the exit of the MNA and the entrance to the organism through an effective penetration of the external barrier of the skin. Figure 8 and Figure 9 represent the behavior that lidocaine could have as a working fluid from the time it enters the MNA until it reaches the dermis.

The results indicate that the velocity decreased considerably as the organism entered, as shown in Figure 10, which corresponds to a velocity profile at the transition from the MNA and the porous media. This is due to the inlet pressure, which is 5 times the pore resistance pressure, defined as resistive pressure in Figure 1. It can also be seen that the flow follows a uniform behavior throughout the system as a whole. It should be noted that in the VF design, the distribution of the fluid towards the capillaries is better, since Figure 9 demonstrates a greater infiltration depth.

From the previous results, it is apparent that the VF design maintains a more uniform velocity distribution and better behavior against resistive interstitial pressure, since the analgesic flow reaches a greater depth towards the capillaries located in the dermis. On the other hand, the LF design, under the same simulation conditions, presents a much more superficial behavior in the delivery of the drug, which may be mainly due to the influence of the resistive pressure and the fact that the delivery profile is not homogeneous along the MNA. Due to the above, it is considered that the pressure of any fluid entering the system must be at least 5 times greater than the resistive pressure of the organism.

Moreover, these results provide a basis for the optimization and understanding of the possible behavior of drugs when entering the organism in the manufactured and/or commercial equipment [30,31].

## 4. Conclusions

The study of the behavior of lidocaine as a working fluid varies greatly depending on the proposed microchannel geometries. These geometric variations result in noticeable differences in the velocity and pressure profile for the LF and VF designs. Accordingly, it is observed that in a complex situation, the transition of the fluid from the MNA and the organism can be affected and result in a more or less pronounced delivery of the drug, which can infer a greater or lesser efficiency in the action of the drug.

It is necessary to point out that, when evaluating the complete transition between the MNA and the porous media, more complex behaviors are obtained, where the input and output parameters condition affect the fluid flow excessively, thus producing an overly idealized situation, where the transcellular pathways allow a fully developed flow.

Moreover, it was shown that a stable model was achieved with an inlet pressure in the MNA that of at least 5 times greater than the resistive pressure of the organism. Then, if the purpose is to model a zone that maintains a higher resistive pressure such as inflammation of the tissue, damaged tissue, presence of cancer cells, or another medical condition that raises the transcellular pressure, the inlet pressure to the system must be increased under the relationship mentioned above.

In addition, due to the application conditions in MN-based devices, the transition between output from the MN array and input into the body is not very well developed. For this reason, a joint simulation is necessary in order to show what types of behaviors of flow could have in these physical situations.

Finally, it should be noted that the handling of parameters such as velocity and pressure required for the application of MNA-based systems should be studied to avoid turbulence at drug entry, which could result in failure to enter the organism, where, in addition, the calibration of the key parameters should be carry out via an experimental validation of the computational results.

## Figures and Tables

**Figure 1 pharmaceutics-15-00358-f001:**
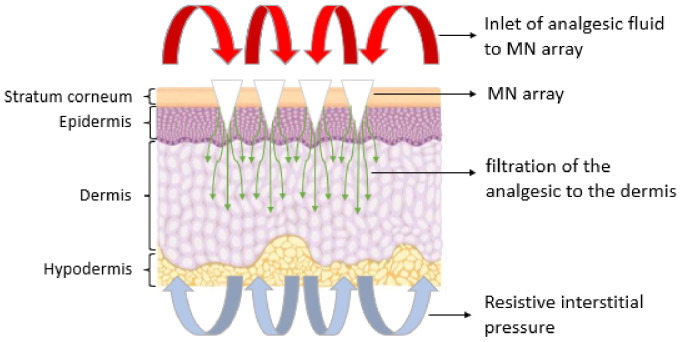
Schematic representation of the physical situation to be simulated.

**Figure 2 pharmaceutics-15-00358-f002:**
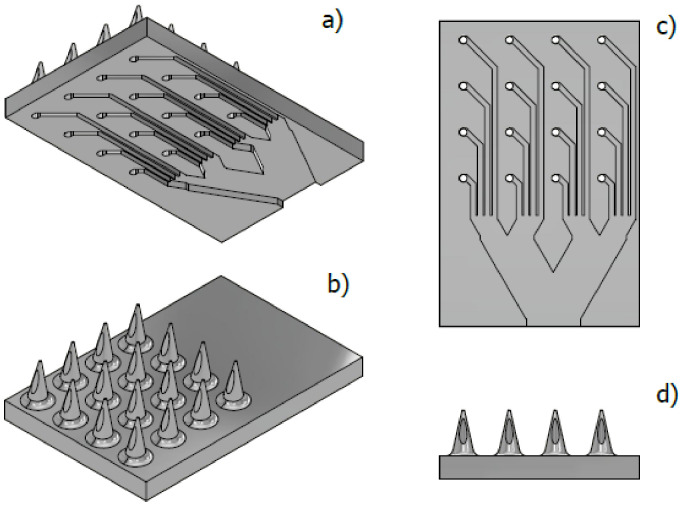
Referential views for the MN 4 × 4 matrix design together with its microchannels. (**a**) Bottom isometric view, (**b**) top isometric view, (**c**) bottom view of microchannels, (**d**) side view of microneedles.

**Figure 3 pharmaceutics-15-00358-f003:**
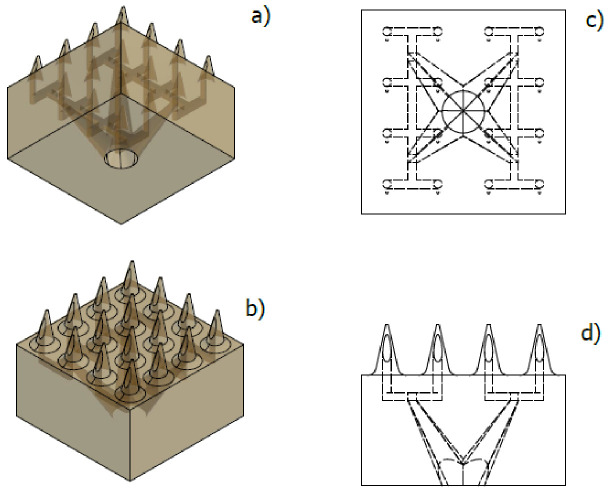
Views for VF design in a MN 4 × 4 array. (**a**) Bottom symmetrical view, (**b**) top isometric view, (**c**) bottom view, (**d**) side view of the MN.

**Figure 4 pharmaceutics-15-00358-f004:**
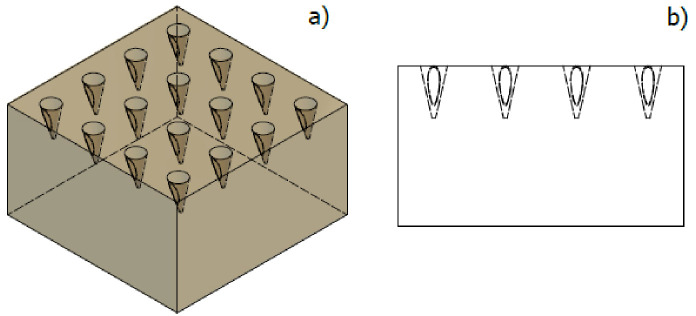
Geometrical design for effective penetration of the 4 × 4 MN array into the epidermis. (**a**) Isometric view, (**b**) side view.

**Figure 5 pharmaceutics-15-00358-f005:**
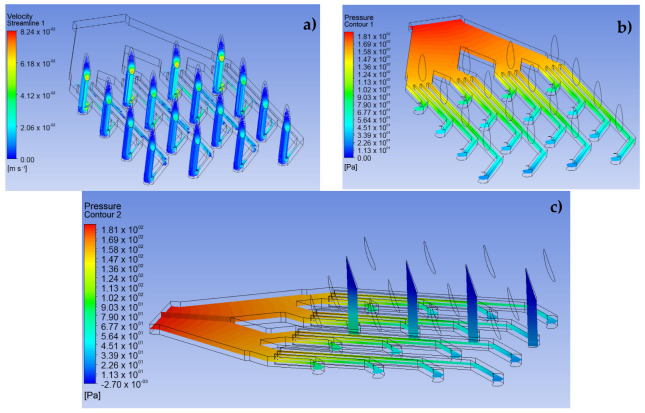
CFD simulation results for design LF in a 4 × 4 array. (**a**) Velocity profile, (**b**) pressure
profile, (**c**) side view of the pressure profile.

**Figure 6 pharmaceutics-15-00358-f006:**
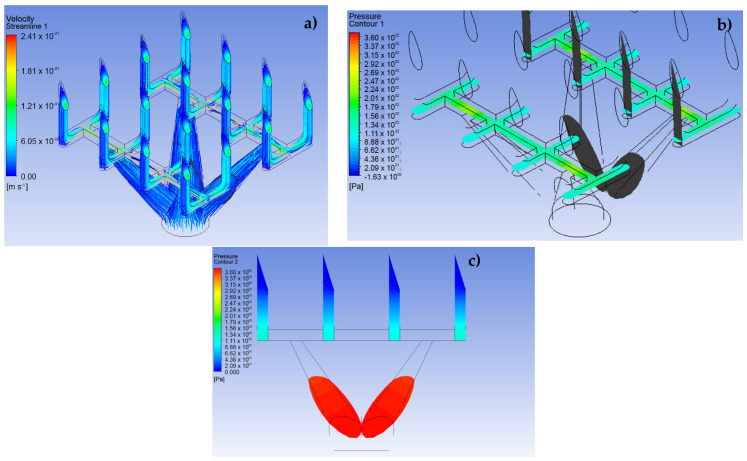
CFD simulation results for design VF in a 4 × 4 array. (**a**) Velocity profile, (**b**) pressure
profile, (**c**) pressure profile for the fluid path.

**Figure 7 pharmaceutics-15-00358-f007:**
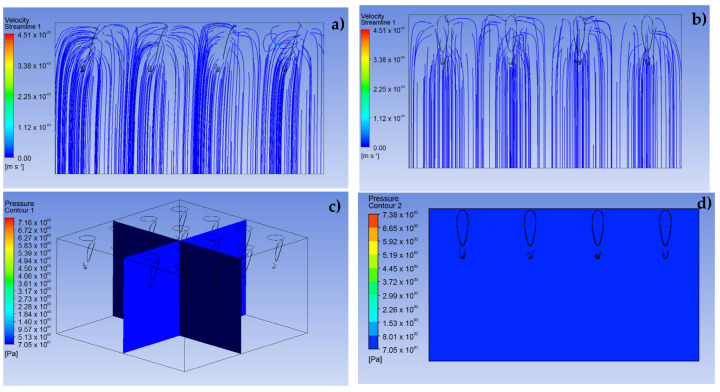
CFD simulation results for the dermis as a porous medium. (**a**) Front view of dermal entry
velocity lines, (**b**) side view of dermis entry velocity lines, (**c**,**d**) isometric view of pressure profile
for fluid entering the dermis.

**Figure 8 pharmaceutics-15-00358-f008:**
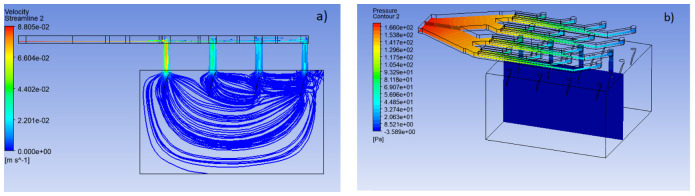
Full CFD simulation of the analgesic delivery system. (**a**) Velocity profile from the MNA
to the skin, (**b**) pressure profile in the MNA.

**Figure 9 pharmaceutics-15-00358-f009:**
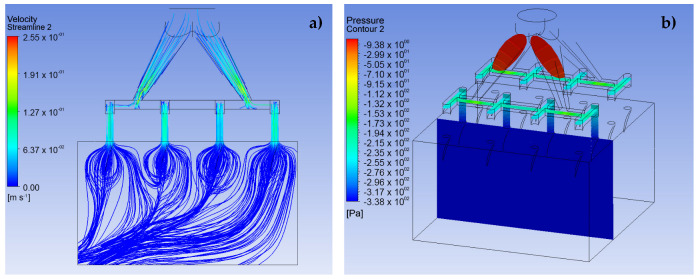
CFD simulation for analgesic administration join system. (**a**) Velocity profile from the
MNA to the skin, (**b**) velocity profile from the MNA to the skin.

**Figure 10 pharmaceutics-15-00358-f010:**
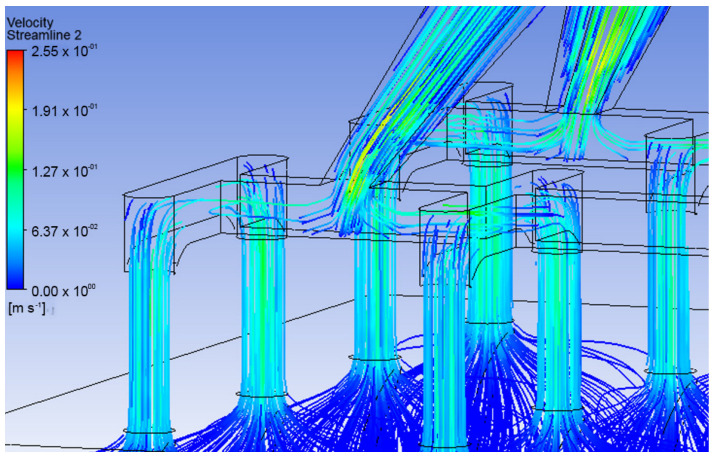
Speed profile view in the VF design.

**Table 1 pharmaceutics-15-00358-t001:** Drugs, antigens, or proteins used in different types of MN arrays for clinical uses.

MN Type	Drugs Used	References
Coating	human erythropoietin alfa, desmopressin, antigenic proteins, inactivated influenza viral proteins, monoclonal IgG, cisplatin, retinoic acid, rapamycin	[3,5,6]
Dissolving	insulin, dextran, monoclonal IgG, recombinant HIV antigens, chondroitin sulfate, B16F10 melanoma with melanin, antigen OVA, antigen adeno virus, antigen TT/DT	[3,5,7]
Hollow	lidocaine, ibuprofen, insulin, S-OIV H1N1, influenza vaccine, glucagon	[3,9,10]

## Data Availability

The data that support the findings of this study are available from the corresponding authors upon reasonable request.

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
