# Peer review of "Modeling of Microneedle Arrays in Transdermal Drug Delivery Applications"

_pharmaceutics, 2023, doi:10.3390/pharmaceutics15020358_

Round 1

Reviewer 1 Report

The authors present the results on the behavior of lidocaine, an analgesic fluid, through an MNA and determine how the liquid would be distributed through the transcellular pathways of the skin using CFD simulation tools.

Several text errors in the manuscript must be corrected, such as a lack of references: page 2, second paragraph, page 3, Microneedle designs, first paragraph.

In the numerical simulation section in the second stage, how is the non-return of the fluid justified if there is a probability that it will cause an MN if there is one due to the specifically located insertion point of the skin? It is not clear if the condition should justify better.

What is the time window to complete the dosage given the MN number per array?

Reviewer 2 Report

Remove word “CFD” from title.

Keywords should be sorted alphabetically

Authors comment on the advantage and disadvantages of MN.

The figure legends need improvement. All legends should have enough description for a reader to understand the figure without having to refer back to the main text of the manuscript.

Typographic and grammatical errors make reading and interpretation of this article difficult. Authors should at the minimum proof read the entire manuscript for typographical errors and fix all grammatical errors.

Table caption should have enough description for a reader to understand the table without having to refer back to the main text of the manuscript.

All acronyms for national agencies, examinations, etc., should be spelled out the first time they are introduced in text or references. Thereafter the acronym can be used if appropriate, e.g. “The work of the Assessment of Performance Unit (APU) in the early 1980s …” and subsequently, “The APU studies of achievement …”, in a reference “(Department of Education and Science [DES] 1989a)”.

Please correct this error throughout the text (Error! Reference source not found.)

Reviewer 3 Report

The manuscript titled CFD modeling for microneedle arrays in transdermal drug delivery applications, in its current form, requires major revisions before it is accepted for publication.

-Written English needs to be improved. Kindly refresh the manuscript for sentence structure and grammatical errors

-Refresh the entire materials and methods section. Add a materials section where you list down all the material that was used in this manuscript along with the vendor/source of the materials. Followed by this, add a methods section and elaborate the methods that were used for each experiment in detail; such that the reader will have enough guidance if they want to reproduce the experiment

-Add a discussion section and discuss in brief about similar studies and how your study stands out in comparison to those that are published

-Since this publication is intended for the journal Pharmaceutics, I would have liked to see at least some MN formulation that was fabricated using the knowledge of CFD. It would be interesting to compare if your outcomes from CFD simulation are reproducible practically

-Since this is a simulation based paper, I would like to see multiple types of drugs being simulated using CFD, not just lidocaine. Lidocaine is bcs class 3. Ideally, the manuscript should cover drugs from all BCS classes

-Drugs entrapped in MN would, in most cases, never be administered without a polymeric backbone. Can CFD involve studying permeation of the drug solution dispersed in a polymeric matrix? If so, kindly add this part. Also, you may add this as a part of discussion

-Kindly mention in discussion about the reliability and reproducibility of CFD in designing MN products commercially

-All but 2 references are from before 2020. Update the manuscript to have more recent references from the past 2 years

Round 2

Reviewer 3 Report

Rebuttal is acceptable